# Optimal utilization of prevention of mother-to-child transmission of HIV services among adolescents under group versus focused antenatal care in Eastern Uganda

**Rebecca Akunzirwe** [1]*, **Sabrina Bakeera-Kitaka**[2], **Joan N. Kalyango**[1], **Jane Frances Zalwango**[1], **Judith Amutuhaire Ssemasaazi**[1], **Tom Okello**[1], **Remmy Buhuguru**[1], **Sarah Kiguli**[2], **Aloysius G. Mubuuke**[3], **Sam Ononge**[4]

1 Clinical Epidemiology Unit, College of Health Sciences, Makerere University, Kampala, Uganda,
2 Department of Pediatrics and Child Health College of Health Sciences, Makerere University, Kampala, Uganda, 3 Department of Radiology, School of Medicine, Makerere University, Kampala, Uganda, 4 Department of Obstetrics and Gynecology, College of Health Sciences, Makerere University, Kampala, Uganda

* akunzirwe@gmail.com

## Abstract

### Background

Group antenatal care (G-ANC), an alternative to focused ANC (F-ANC), involves grouping mothers by gestational and maternal age. In high-income countries, G-ANC has been associated with improved utilization of health care services like Prevention of Mother to Child Transmission (PMTCT) of HIV services. Some low-resource countries with poor utilization of health care services have piloted G-ANC. However, there is limited evidence of its efficiency. We, therefore, compared G-ANC versus F-ANC with regards to optimal utilization of PMTCT of HIV services and assessed associated factors thereof among adolescent mothers in eastern Uganda. We defined optimal utilization of PMTCT of HIV services as the adolescent being up to date with HIV counseling and testing. If found HIV negative, subsequent timely re-testing. If found HIV positive, initiation of antiretroviral therapy (ART) under option B plus for the mother. While for the infant, it entailed safe delivery, testing, feeding, and appropriate HIV chemotherapy.

### Methods

From February to April 2020, we conducted a cross-sectional study among 528 adolescent mothers in four sites in eastern Uganda. We assessed the optimal utilization of PMTCT of HIV services among adolescent mothers that had attended G-ANC versus F-ANC at the post-natal care or immunization clinics. We also assessed the factors associated with optimal utilization of PMTCT of HIV services among these mothers.

### Results

Optimal utilization of PMTCT was higher among those in G-ANC than in F-ANC (74.7% vs 41.2, p-0.0162). There was a statistically significant association between having attended

**Data Availability Statement:** This minimal dataset for replicating findings from this study can be found at https://doi.org/10.5281/zenodo.6511239.

**Funding:** This research was supported by the Forgarty International Centre of the National Institutes of Health, U.S. Global AIDS Coordinator and Health Diplomacy (S/GAC), and President's Emergency Plan for AIDS Relief (PEPFAR) under Award Number 1R25TW011213. The research award was received by AR. The funders had no role in study design, data collection and analysis, decision to publish, or preparation of the manuscript.

**Competing interests:** The authors have declared that no competing interests exist.

G-ANC and optimal utilization of PMTCT [PR = 1.080, 95%CI (1.067–1.093)]. Other factors independently associated with optimal utilization were; having a partner that tested for HIV [PR = 1.075, 95%CI (1.048–1.103)], trimester of first ANC visit: second trimester [PR = 0.929, 95%CI (0.902–0.957)] and third trimester [PR = 0.725, 95%CI (0.616–0.853)], and the health facility attended: Bugembe HCIV [PR = 1.126, 95%CI (1.113–1.139)] and Jinja regional referral hospital [PR = 0.851, 95%CI (0.841–0.861)]

## Conclusions

Adolescent mothers under G-ANC had significantly higher optimal utilization of PMTCT of HIV services compared to those under F-ANC. We recommend that the Ministry of Health considers widely implementing G-ANC, especially for adolescent mothers.

## Introduction

Globally, adolescent pregnancies have remained unacceptably high despite many efforts to reduce them [1]. In 2020, 41 per 1000 girls aged 15 to 19 years gave birth globally. Africa had the highest birth rate at 98 per 1000 girls and Uganda was ranked 12th among countries with the highest rate of adolescent births at 113 births per 1000 girls [2].

Ironically, countries with high adolescent pregnancies like Uganda have also shown a high rate of HIV incidence [3]. This predisposes the adolescents to a high risk of HIV acquisition and ultimately vertical transmission to their unborn infants [4, 5]. Mother-to-child transmission (MTCT) of HIV is one of the main HIV transmission modes, with MTCT of HIV rates ranging from 15-to 45% [6, 7]. This rate, however, can be reduced to less than 5% with effective interventions to prevent transmission of HIV from mother to child (PMTCT) services [6].

The PMTCT strategy is a cascade of events that enable the identification of an HIV-positive pregnant woman and the prevention of HIV transmission to her infant. It comprises attending antenatal care (ANC), HIV counseling and testing (HCT), and receiving HIV test results for the mother. If negative, subsequent retesting should be done in the third trimester, shortly after delivery, or within three months of the last test. If positive, ART should be given for life to the mother under option B plus, while exposed infants should be given Nevirapine syrup (NVP) and cotrimoxazole prophylaxis, and, if found positive, anti-retroviral therapy (ART). In addition, the strategy supports safe delivery (delivery in a health facility by a qualified health worker) and safe infant feeding (exclusive breastfeeding for six months, complementary feeding for up to 12 months for HIV-exposed infant and 24 months for HIV positive infants) with early infant diagnosis within six to eight weeks of birth. DNA PCR is done six weeks after ceasing breastfeeding, serology at 18 months and if DNA PCR is positive, they are linked to care [6].

ANC is crucial for the utilization of PMTCT services as it may provide the first opportunity for HIV Counselling and Testing as well as an entry point for care and support for HIV-positive pregnant women, their partners, and newborn babies [6].

In Uganda, focused antenatal care (F-ANC), a form of individualized care between a pregnant woman and a health worker is recommended by the Ministry of Health [8]. From the available literature, there are several challenges in the delivery and subsequent utilization of PMTCT services under F-ANC, like the ineffective organization of educational sessions, selective omission of certain services, lack of explanation of important clinical and laboratory

procedures, and occasional lack of respect for clients [9–11]. Group ANC (G-ANC), which entails smaller groups of women (8 to 10) with similar maternal and gestational age, and a health worker has been suggested as an alternative to F-ANC. It integrates individualized pregnancy health assessment with group educational activities and peer support according to the women's needs [12]. Some of the reported benefits of G-ANC include; improved health literacy, ANC adherence, improved quality of care for pregnant women, and enhanced utilization of PMTCT services [13]. Although G-ANC, which is being piloted in Uganda under a four visit model, is feasible and acceptable [14], factors associated with the utilization of PMTCT services among adolescent pregnant women remain scarce. The study aimed to compare the level of utilization of PMTCT among those receiving ANC service under G-ANC to those under F-ANC.

## Materials and methods

### Study design and setting

This was a cross-sectional study from February to April 2020 in health facilities in the Busoga region, eastern Uganda. The rate of teenage pregnancy was 20.7% in 2016 while that of HIV prevalence was 5.1% in the eastern region in 2020 [15, 16]. There are 31 hospitals and 48 health center fours [HCIVs- (mini hospital headed by a senior medical officer at a sub-district level)] in eastern Uganda [17]. During the study period, only one district hospital and three HCIVs offered G-ANC. We purposely chose four facilities to conduct the study because they had a large catchment area. Two facilities (Iganga hospital and Bugembe health center IV) offered G-ANC while Jinja regional referral hospital and Budondo HCIV offered F-ANC. In addition, the selected facilities offered immunization, post-natal services, PMTCT services, early infant diagnosis, and youth-friendly services among others.

Some mothers visited the postnatal clinic up to six weeks post-partum. Mothers brought the infants for immunization services when they were six weeks, 10 weeks, 14 weeks, and nine months old. Facility data from the selected study sites indicate that immunization clinics received 30–40 babies per working day and between five to twenty mothers were attended to at the postnatal clinic. Of these mothers, an average of two to seven mothers were adolescents. The clinics were staffed by one to two- midwives.

### Study participants and sampling

We consecutively enrolled 528 adolescent mothers that had sought immunization or postnatal services from the four health facilities in eastern Uganda. Mothers were included if they met all the following criteria: a) sought these services during the study period b) had completed at least 18 months of care if HIV positive, and 9 months of care if HIV negative (to capture mothers at the end of the PMTCT cascade), c) had alive infants 9 months or less d) provided written informed consent. We excluded those: a) unable to withstand the interview because they or their infants were too ill, b) who had attended both focused ANC as well as group ANC.

### Sample size

The sample size was estimated using the formula for comparison of two proportions [18]. We assumed a 5% level of significance and 80% power. The sample size was calculated to detect a small absolute effect size of 0.2 [19]. The proportion of mothers under F-ANC that optimally utilized these services was 0.3 [20], thus the proportion that was anticipated to optimally utilize these services under G-ANC was 0.5; giving a total of 206 participants. We further adjusted

this sample size by a design effect of 2 to cater for anticipated clustering at the health facility level and also catered for a possible non-response of 10%. Thus, a sample size of 458 participants was obtained. However, we obtained 528 participants to improve power to detect differences in other associations with optimal utilization of PMTCT of HIV services.

## Variables

The outcome variable, optimal utilization of PMTCT of HIV services was defined as being up to date with HIV counseling and testing (HCT) (that is the adolescent had HCT and received test results at any point during the pregnancy but before the third trimester. If the result was negative subsequent retesting was done in the third trimester, at delivery, shortly after delivery (up to six weeks after delivery), or after three months of the previous HIV test. If the mother was positive she was initiated on ART under option B plus. In addition, there was safe infant delivery, safe infant feeding, infant testing, and receipt of results, nevirapine was given for six weeks, and cotrimoxazole was given until status was confirmed. Infant testing entailed early infant diagnosis (DNA PCR within 6–8 weeks), if the test turned positive, ART was initiated and the infant was linked into care.

We obtained data on the main exposure variable, the mode of ANC that is whether G-ANC or F-ANC. This was assessed by the health facility (Bugembe HCIV and Iganga hospital offered G-ANC while JRRH and Budondo HCIV offered F-ANC). It was also assessed by mode written on the ANC cards and by asking the mothers whether they had received ANC care from any other health facilities. We also obtained data on socio-demographic characteristics (marital status, education, parity, religion), household, and prenatal history (the number of ANC visits, trimester of first ANC contact, and mean clinic travel time). Clinic travel time was determined by asking the mothers how long it took them to get to the health facility from where they received ANC. Lastly, we obtained data on HIV testing characteristics (the type of counseling and testing, disclosure status, and partner testing) and PMTCT utilization characteristics (HCT status, trimester of testing, re-testing, and receipt of results, if positive: whether on ART, infant delivery in hospital, type and length of breastfeeding, infant DNA PCR test status, nevirapine within 6 weeks, whether on cotrimoxazole, ART for HIV positive infant.

## Data collection

We trained 6 interviewers (diploma-level nurses) for 4 days on the questionnaire, data collection procedures, and sampling methods. We pretested a translated questionnaire on 5 adolescent mothers who were excluded from the study. Mothers arriving at the clinics for a needed service were screened after they had received the service and assessed to ascertain whether they met the eligibility criteria. Adolescent mothers who met the eligibility criteria were approached and the purpose of the study was explained to them.

Mothers who accepted to participate in this study were consented, enrolled in the study, and given a study identification number. The interviewer-administered questionnaire was used to obtain information on the participants' social demographic characteristics, HIV testing as well as PMTCT of HIV service utilization.

The information on utilization of PMTCT of HIV services was cross-referenced with that in their antenatal cards, the ANC, or the PMTCT register.

## Data management & statistical analysis

Data were double entered into Epidata version 4.1.1.0 (EpiData Association, Odense, Denmark) and exported to Stata version 14.0 (Stata Corporation, College Station, Texas, USA). for analysis. Data were declared as survey data to adjust for clustering by study site. A stratified

analysis based on the mode of antenatal care was carried out for descriptive analysis of baseline characteristics of the study population and reported as frequencies, proportions, means, and medians.

Due to there being a small number of balanced clusters, the optimal utilization of PMTCT of HIV services among adolescents under G-ANC versus F-ANC was compared using cluster adjusted t-statistic [21].

To assess for: a) the relationship between each component of the PMTCT cascade and ANC mode and b) the associations between optimal utilization of PMTCT services and possible associated factors; a generalized linear fixed effects model using the Poisson family and log link reporting robust standard errors was used. The reported measure of association was prevalence ratios with their 95% confidence intervals. Prevalence ratios between individual predictors and the optimal utilization of PMTCT of HIV services at the bivariate level were assessed and those with a p-value less than or equal to 0.2 were considered for the multivariable analysis. A p-value less than or equal to 0.05 was considered statistically significant. The presence of interaction was assessed by using the log-likelihood test comparing the full and reduced models. Confounding variables were assessed by comparing prevalence ratios of unadjusted and adjusted models and those that caused a difference of at least 10% were considered confounders. The goodness of fit of the regression model to the data was assessed using the deviance test.

## Ethical approval

Ethical approval was obtained from the Makerere University College of Health Sciences School of Medicine Research Ethics Committee (SOMREC) #REC REF 2020–023. Administrative permission was obtained from the 4 health facilities. Written informed consent was obtained from study participants.

## Results

### Characteristics of the study participants

Of the 572 adolescent mothers screened, 528 (92.31%) were enrolled. Of the 44 mothers that were not enrolled, 33 had children older than nine months, and 11 declined to consent to participate in the study. Budondo HCIV and Iganga hospital, each contributed 88 mothers while Bugembe HCIV and Jinja regional referral hospital contributed 132 participants each. The minimum and maximum participant ages were 14.0 and 19.0 years respectively. The majority of the adolescents (82.6%) were 18 years old and over. Participants under G-ANC had more ANC visits compared to those under F-ANC (Table 1).

**Comparison of optimal utilization of PMTCT services among adolescents by ANC groups.** There was a higher mean proportion of adolescents under G-ANC that optimally utilized PMTCT of HIV services (74.7%) compared to those under F-ANC (41.2%) The difference between the means of proportions in the two modes of ANC was 33.5% (95% CI 14.9%-52.1%**, p-0.0162)**

**Utilization of the different components of PMTCT cascade among HIV-negative adolescent mothers in Eastern Uganda.** Among the HIV-negative adolescent mothers, 257 were under G-ANC while 252 were under F-ANC. One of the mothers under G-ANC reported never having HCT. Ninety-one (39.7%) of the mothers under F-ANC that had up-to-date HCT had timely re-testing compared to 203(78.9%) for those under G-ANC. Seven of the mothers under F-ANC that had timely re-testing were not up to date with the first HCT (Table 2).

**Table 1. Characteristics of 528 adolescent mothers enrolled in the study in eastern Uganda.**

| Characteristic | G-ANC | F-ANC |
|---|---:|---:|
| | N (%) | N (%) |
| Number of participants | 264 | 264 |
| Median age (1st, 3rd quartile) | 19 (18,19) | 19 (18,19) |
| **Marital status** | | |
| Single/ widowed | 52 (19.7) | 70 (26.5) |
| Married /cohabiting | 205 (77.7) | 170 (64.4) |
| Separated/Divorced | 7 (2.7) | 24 (9.1) |
| **Education** | | |
| None | 3 (1.1) | 8 (3.03) |
| Primary | 99 (37.5) | 109 (41.3) |
| Secondary | 145 (54.9) | 133 (50.4) |
| Tertiary | 17 (6.4) | 14 (5.3) |
| **No. of Visits** | | |
| <4 | 19 (7.2) | 142 (53.8) |
| ≥4 | 245 (92.8) | 122 (46.2) |
| **Religion** | | |
| Christian | 149 (56.4) | 171 (64.8) |
| Moslem | 115 (43.6) | 93 (35.2) |
| **First ANC visit** | | |
| First trimester | 110 (41.7) | 111 (42.1) |
| Second trimester | 145 (54.9) | 136 (51.5) |
| Third trimester | 9 (3.4) | 17 (6.4) |
| **Type of counseling and testing** | | |
| VCT | 63 (23.9) | 64 (24.2) |
| Provider initiated counseling and testing | 200 (76.1) | 200 (75.8) |
| **HIV test result** | | |
| Positive | 6 (2.3) | 12 (4.5) |
| Negative | 257 (97.7) | 252 (95.5) |
| **Parity** | | |
| 1 child | 244 (92.4) | 200 (75.8) |
| >1 child | 20 (7.6) | 64 (24.2) |
| **Mean travel time to the clinic in minutes (±SD)** | 32.26± 23.32 | 44.44± 26.80 |

P-values were calculated from the Pearson's Chi-square test for categorical predictors and the Wilcoxon rank-sum test for continuous ones.

**Utilization of the different components of PMTCT cascade among HIV-positive adolescent mothers in Eastern Uganda.** Five of the six HIV-positive mothers under G-ANC optimally utilized PMTCT of HIV services. For mothers under F-ANC, 1 of the mothers who reported irregular taking of the anti-retroviral therapy also hadn't had her infant started on cotrimoxazole prophylaxis. Seven of the twelve mothers under F-ANC optimally utilized PMTCT of HIV services (Fig 1)

## Factors associated with optimal utilization of PMTCT services among adolescents in eastern Uganda

At multivariable analysis, we did not find HIV status to be statistically significant. However, having: attended G-ANC [PR = 1.080, 95%CI (1.067–1.093)], partner that tested for HIV

**Table 2. Optimal utilization of PMTCT services for 509 HIV-negative adolescent mothers in Eastern Uganda.**

| Step in PMTCT cascade | G-ANC, n (%) | F-ANC, n (%) | cPR | 95% CI | P-value |
|---|---|---|---|---|---|
| **HIV negative** | 257 | 252 | N/A | N/A | N/A |
| Up to date with 1ˢᵗ HCT | 247 (96.1) | 210 (83.3) | 1.070 | 1.022–1.200 | 0.004 |
| Receipt of results | 257 (100.0) | 252 (100.0) | 1.000 | N/A | N/A |
| Timely re-testing | 203 (78.9) | 98 (38.9) | 2.031 | 1.472–2.803 | <0.001 |
| Optimal utilization of PMTCT of HIV services | 203 (78.9) | 91 (39.7) | 1.315 | 1.150–1.503 | <0.001 |

cPR (crude prevalence ratio) comparing G-ANC to F-ANC, F-ANC reference category. cPR are computed from modified Poisson regression with robust standard errors.

[PR = 1.075, 95%CI (1.048–1.103)], trimester of first ANC: second trimester [PR = 0.929, 95% CI (0.902–0.957)], third trimester [PR = 0.725, 95%CI (0.616–0.853)], and the health facility attended: Bugembe HCIV [PR = 1.126, 95%CI (1.113–1.139)] and Jinja regional referral hospital [PR = 0.851, 95%CI (0.841–0.8610)] (Table 3).

In the bivariate analysis: being younger [PR = 1.090, 95%CI (1.042–1.141)], clinic travel time per minute increase [PR = 0.998, 95%CI (0.997–0.999)], attending ≥4 ANC visits [PR = 1.304, 95%CI (1.087–1.563)], and having more than one child [PR = 0.863, 95%CI (0.771–0.965)] were associated with optimal utilization of PMTCT of HIV services but these associations were not significant at the multivariable analysis (Table 3).

## Discussion

This study compared the optimal utilization of PMTCT of HIV services among adolescents under G-ANC versus F-ANC in Eastern Uganda. We also assessed for factors associated with optimal utilization of PMTCT of HIV services among these adolescents. The mean optimal utilization of PMTCT of HIV services was 74.7% under G-ANC and 41.2% under F-ANC. The

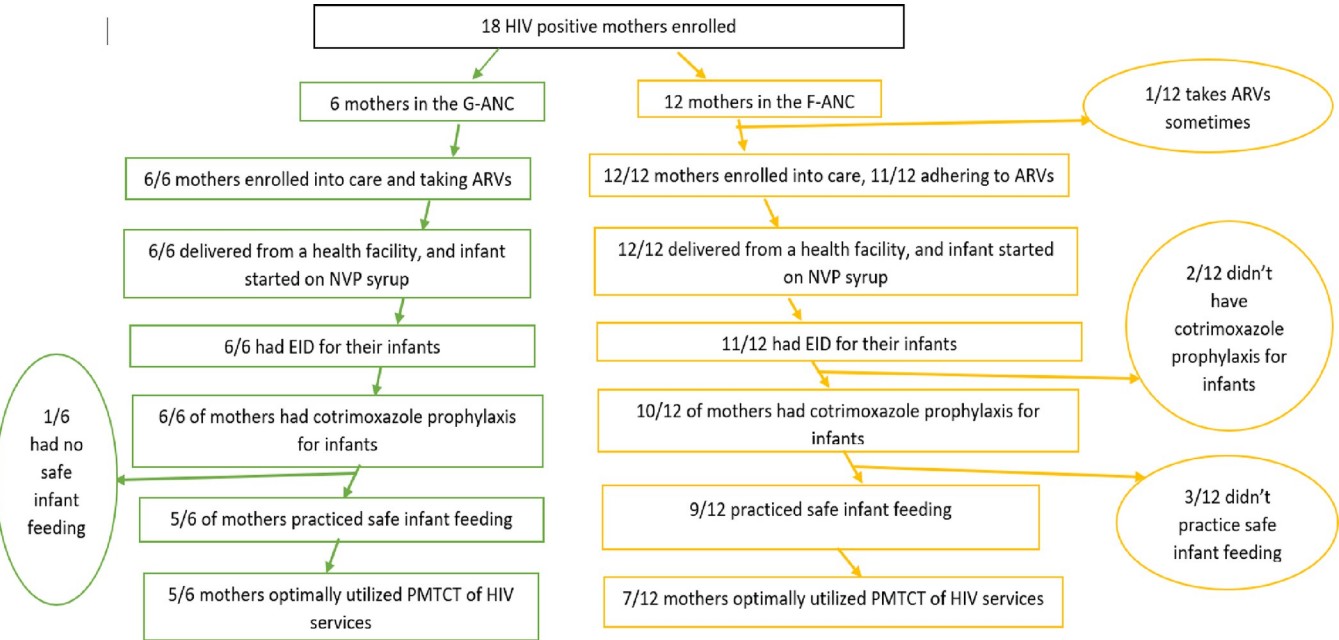

**Fig 1. PMTCT cascade utilization for 18 HIV-positive mothers in Eastern Uganda.** F-ANC G-ANC.

**Table 3. Factors associated with optimal utilization of PMTCT of HIV services among 528 adolescent mothers in Eastern Uganda.**

| Variable | Categories | Optimal utilization of PMTCT services (%) | Unadjusted analysis | | Adjusted analysis | |
|---|---|---|---|---|---|---|
| | | | PR (95% CI) | P-value | PR (95% CI) | P-value |
| Type of ANC | F-ANC | 98/264 (37.1%) | reference group | | reference group | |
| | G-ANC | 208/264 (78.8%) | 1.304 (1.141–1.490) | <0.001 | 1.080 (1.067–1.093) | <0.001 |
| HIV test result | negative | 294/509 (57.8%) | reference group | | reference group | |
| | positive | 12/18 (66.7%) | 1.056 (0.951–1.173) | 0.304 | 1.092 (0.992–1.203) | 0.072 |
| Age | > = 18 years | 242/436 (55.5%) | reference group | | reference group | |
| | <18 years | 64/92 (69.6%) | 1.090 (1.042–1.141) | <0.001 | 1.044 (0.988–1.104) | 0.123 |
| Clinic travel time (per min increase) | | N/A | 0.998(0.997–0.999) 0.006 | | 0.999(0.999–1.000) 0.605 | |
| Education | None/primary | 115/219 (52.5%) | reference group | | reference group | |
| | Secondary/tertiary | 191/309 (61.8%) | 1.061 (0.991–1.136) | 0.088 | 1.032 (0.982–1.085) | 0.216 |
| No. of ANC visits | <4 | 49/161 (30.4%) | reference group | | reference group | |
| | ≥4 | 257/367 (70.0%) | 1.304 (1.087–1.563) | 0.004 | 1.123 (0.952–1.324) | 0.170 |
| Health facility | Iganga GH | 55/88 (62.50%) | reference group | | reference group | |
| | Jinja RRH | 55/176 (31.25%) | 0.808 (0.745–0.876) <0.001 | | 0.851 (0.841–0.861) <0.001 | |
| | Bugembe HCIV | 153/176 (86.93%) | 1.150 (1.075–1.231) <0.001 | | 1.126 (1.113–1.139) <0.001 | |
| | Budondo HCIV | 43/88 (48.86%) | 0.916 (0.834–1.006) | 0.067 | - | - |
| First ANC visit | First trimester | 149/221 (67.4%) | reference group | | reference group | |
| | Second trimester | 152/281 (54.1%) | 0.929 (0.889–0.952) | <0.001 | 0.929 (0.902–0.957) | <0.001 |
| | Third trimester | 5/26 (19.2%) | 0.712 (0.604–0.840) | <0.001 | 0.725 (0.616–0.853) | <0.001 |
| Partner tested | No/ I don't know | 98/264 (37.1%) | reference group | | reference group | |
| | Yes | 208/264 (78.8%) | 1.304 (1.141–1.490) | <0.001 | 1.075 (1.048–1.103) | <0.001 |
| Number of children | 1 | 273/444 (61.5%) | reference group | | reference group | |
| | >1 | 33/84 (39.3%) | 0.863 (0.771–0.965) | 0.010 | 0.961 (0.894–1.034) | 0.286 |

PMTCT (Prevention of Mother to Child Transmission of HIV), PR-Prevalence ratio, ANC (Antenatal Care), HCIV- Health Center IV

factors that were found to be significantly associated with optimal utilization of PMTCT of HIV services are: having attended G-ANC, partner having tested for HIV, Health facility attended, and earlier trimester of first ANC visit.

On average, 75 per 100 adolescents under G-ANC had optimally utilized PMTCT of HIV services compared to 42 per 100 of those under F-ANC with a difference of 33.5% between the two groups. This difference was statistically significant with a p-value of 0.0162. Participants under G-ANC were 8% more likely to have optimally utilized PMTCT services compared to those under F-ANC.

The observed better optimal utilization of PMTCT of HIV services under G-ANC could be attributed to improved client satisfaction which improves health care utilization. This is due to the participatory nature of G-ANC which improves relationships between health care workers and mothers as well as among the mothers [22–25]. Our findings of the mean optimal utilization of PMTCT services of 41.2% among adolescent mothers under F-ANC was higher than that of a cross-sectional study carried out at Mulago national referral hospital where optimal utilization of PMTCT of HIV services was reported to be 30% [20]. This could be because the eastern region has shown a higher utilization of PMTCT services than the central region of Uganda [26]. This is probably because of the intensified efforts to promote adolescent maternal and child health in this region by both the ministry of health and implementing partners compared to the central region. These include the Regional health integration to enhance services in the Busoga region [27], the government of Korea's partnership with WHO, and the Ministry of Health to support maternal and child health delivery in Busoga [28] among others [29, 30].

As such, there has been collaboration with village health teams for mapping pregnant adolescents; who are then linked to health care. The observed difference could also be because the findings in central Uganda were from a study carried out in a national referral hospital. Therefore, there was likely difficulty in follow-up and as such poorer PMTCT service utilization. Optimal utilization of PMTCT of HIV services was also low among adolescents under G-ANC compared to the national target of 90% of e-MTCT retention [31]. It is however a positive step toward the 'All In initiative' to end adolescent AIDS, as well as the global health sector strategy of elimination of mother-to-child transmission of HIV by 2030 [32, 33]. Though literature is scarce on the utilization of PMTCT of HIV services in the context of G-ANC or compared to F-ANC, a prospective cohort of HIV-positive pregnant women found that there were decreased missed ARV doses in the pre and post-G-ANC periods [34].

Adolescent mothers that attended their first ANC visit in the second and third trimesters were 7% and 28% less likely to have optimally utilized PMTCT of HIV services than those whose first ANC contact was in the first trimester. This is because earlier ANC means earlier education, detection, and management of danger signs of pregnancy as well as HIV [6, 35]. Our results are similar to those of a study in South Africa where every one-week delay in gestational age of first ANC booking increased early MTCT of HIV by 10% and yet the odds of MTCT among HIV positive adolescents that had any PMTCT intervention was 0.2 [36].

Adolescent mothers that attended Bugembe HCIV and Jinja regional referral hospitals were 13% more likely and 15% less likely respectively to have optimally utilized PMTCT of HIV services than those from Iganga general hospital. This is probably because Bugembe HCIV has a smaller catchment area while Jinja Regional Referral Hospital has a larger catchment area compared to Iganga general hospital [17]. Larger catchment areas with larger populations likely result in high workloads [37, 38] which affects the quality of patient care [39]. Our results are consistent with those of a study where patients in health center settings were 0.358 (0.231–0.555) times as likely to have been lost to follow-up as those in hospitals [40].

Adolescent mothers that had a partner who had tested for HIV were 8% more likely to have optimally utilized PMTCT services than those whose partners either did not test or had an unknown testing status. This could be because the adolescents' partners whose testing status is known, have greater involvement in the PMTCT program. This has been associated with higher PMTCT service utilization. It could also be that these had couple's HIV testing which increases HIV testing for pregnant women, a component of the PMTCT cascade [41]. Our results are consistent with those of a study in Ethiopia in 2015 whereby the odds of utilizing PMTCT services among mothers whose partners had undergone HIV counseling and testing were 8.2 times those of mothers whose partners had not undergone it [42, 43].

HIV test result was not significantly associated with optimal utilization of PMTCT of HIV services. This is in contrast to those reported in a study in Uganda where HIV-positive mothers were 18.2 times as likely to optimally utilize PMTCT of HIV services as HIV-negative mothers [20]. This might be due to our study's small number of HIV-positive mothers.

This study highlights the important role that G-ANC could play in improving the optimal utilization of PMTCT services among adolescent mothers in resource-limited settings. We also believe that our study results are valid given that information on PMTCT service utilization was cross-checked with that in the ANC cards, ANC, and PMTCT registers. Our study was not without limitations. Firstly, this study had a potential for selection bias as it was health center-based, therefore limiting the results to older adolescents who had good health-seeking behavior. These findings can however be generalized to this group of adolescents getting health care from other health facility settings. Secondly, this being a cross-sectional study, it is not possible to establish a causal relationship between the mode of ANC and optimal utilization of HIV services. That said, the groups i.e. F-ANC and G-ANC were similar concerning patient

characteristics thus the observed difference is most probably due to the mode of ANC. Thirdly, the study recruited a small number of clusters than the minimum 30 sites required for a comparative study. However, cluster adjusted T statistic and fixed effects regression modeling were used to adjust for this.

We recommend that taken cautiously, findings from this study support wider implementation of G-ANC by the Ministry of Health especially among adolescents as it appears to be adolescent-friendly in Uganda. We also recommend community sensitization of early attendance of ANC perhaps using community health workers to enable early attendance of ANC. Future researchers should consider the use of prospective research methods to determine the impact of G-ANC on the utilization of PMTCT services.

## Supporting information

**S1 Table.**
(PDF)

## Acknowledgments

We thank miss Evelyn Bakengesa and Mr. Orishaba Philip for the technical support rendered in the conduct of this study. We also thank the study team and participants for contributing to this study.

## Author Contributions

**Conceptualization:** Rebecca Akunzirwe, Sabrina Bakeera-Kitaka, Joan N. Kalyango, Jane Frances Zalwango, Judith Amutuhaire Ssemasaazi, Sarah Kiguli, Aloysius G. Mubuuke, Sam Ononge.

**Data curation:** Rebecca Akunzirwe.

**Formal analysis:** Rebecca Akunzirwe, Sabrina Bakeera-Kitaka, Joan N. Kalyango, Jane Frances Zalwango, Judith Amutuhaire Ssemasaazi, Tom Okello, Remmy Buhuguru, Sam Ononge.

**Funding acquisition:** Rebecca Akunzirwe, Joan N. Kalyango, Sarah Kiguli, Aloysius G. Mubuuke.

**Investigation:** Rebecca Akunzirwe.

**Methodology:** Rebecca Akunzirwe, Sabrina Bakeera-Kitaka, Joan N. Kalyango, Jane Frances Zalwango, Remmy Buhuguru, Sarah Kiguli, Aloysius G. Mubuuke, Sam Ononge.

**Project administration:** Rebecca Akunzirwe, Sam Ononge.

**Supervision:** Rebecca Akunzirwe, Sabrina Bakeera-Kitaka, Joan N. Kalyango, Sarah Kiguli, Aloysius G. Mubuuke, Sam Ononge.

**Validation:** Rebecca Akunzirwe, Tom Okello, Sam Ononge.

**Visualization:** Rebecca Akunzirwe, Sabrina Bakeera-Kitaka, Jane Frances Zalwango, Judith Amutuhaire Ssemasaazi, Tom Okello, Sam Ononge.

**Writing – original draft:** Rebecca Akunzirwe.

**Writing – review & editing:** Rebecca Akunzirwe, Sabrina Bakeera-Kitaka, Joan N. Kalyango, Jane Frances Zalwango, Judith Amutuhaire Ssemasaazi, Tom Okello, Remmy Buhuguru, Sam Ononge.

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
