## [Decision Letter · Decision Letter 0]

22 Mar 2022

PONE-D-21-31253Optimal utilization of Prevention of Mother-to-Child Transmission of HIV services among adolescents under group versus focused antenatal care in Eastern Uganda.PLOS ONE

Dear Dr. Akunzirwe,

Thank you for submitting your manuscript to PLOS ONE. After careful consideration, we feel that it has merit but does not fully meet PLOS ONE’s publication criteria as it currently stands. Therefore, we invite you to submit a revised version of the manuscript that addresses the points raised during the review process.

We look forward to receiving your revised manuscript.

Kind regards,

Orvalho Augusto, MD, MPH

Academic Editor

PLOS ONE

Journal Requirements:

Additional Editor Comments:

This is quite an interesting report on a strategy to improve PMTCT among adolescents. As an alternative to individual antenatal care, the authors are testing a group antenatal care. As outcome, the authors use something they called “optimal utilization of PMTCT of HIV services”. This is a compositive outcome with important components the authors decided not to look at.

Questions/Comments:

There are so many avoidable English typos all over the manuscript. Please, do revise.

Please add in the abstract a short definition of this concept of “optimal utilization of PMTCT of HIV services”.

Line 85 in the study design: the rates presented there are yearly rates? Clarify these figures, please.

The eligibility criteria to be part of the study are quite unclear as the reviewer explains below. Did the women have to fulfil all the criteria in lines 99 to 101? Or just one criteria would be enough to participate in the study?

Sample size: is the effect size an absolute difference of the proportion? Or a relative difference. Clarify.

Line 124: How the exposure was measured here?

Data management and statistical analysis:

- Cite Epidata and Stata

- It is a bit problematic to use a mixed-effects model with just 4 clusters. Even with the sandwiched standard errors. I would suggest fitting fixed-effects models and keeping the health facility as 3 dummy indicators.

- Each one of the components of the “optimal utilization of PMTCT of HIV services” deserves to be analysed. At least show the overall prevalence of these.

Put the Ethical approval within the methods section

There are so many abbreviations. Please provide a list

Results section:

Be consistent with the proportions. In the tables 1 and 2 they are presented as percentages. But on the text lines 171 to 173, for example, they are something else.

Table 1:

- Add the age descriptives described on line 163.

- At the HIV test result. What are the 1 and 2 rows?

- the mean travel time how was measured? This is not described in the methods.

- Please add as footnote how these p-values were computed

For the analysis of optimal utilization in line 169 to 173 I would request to add here the different components of this composite measure. There is an interest to see at what component the intervention did make a change.

Table 2:

- Add one column for prevalence of “optimal utilization of PMTCT services”

- For the first ANC, please change the reference category

- Age in the column of “Yes” we have 242. Add its proportion.

In fact, the analysis of factors here is not that important. It is OK to adjust for these factors to have something like table 3. But I would prefer to have the all composite elements analysed to get their PR for the ANC type.

Table 3 - this is a multivariable analysis. Correct the lone 193 as well. How this model came? How these covariates were chosen?

- Change the reference for the first ANC

Reviewers' comments:

Reviewer's Responses to Questions

**Comments to the Author**

1. Is the manuscript technically sound, and do the data support the conclusions?

Reviewer #1: Partly

2. Has the statistical analysis been performed appropriately and rigorously? 

Reviewer #1: I Don't Know

3. Have the authors made all data underlying the findings in their manuscript fully available?

Reviewer #1: No

4. Is the manuscript presented in an intelligible fashion and written in standard English?

Reviewer #1: No

5. Review Comments to the Author

Reviewer #1: General Comments: This manuscript “ Optimal utilization of Prevention of Mother –to Child Transmission of HIV services among adolescents under group versus focused antenatal care in Eastern Uganda” addresses a timely and challenging topic—how best to ensure uptake of PMTCT services by adolescents. The cross sectional study conducted in postpartum and immunization clinics, reports encouraging results with use of age and gestational grouping of antenatal services (“G-ANC) in terms of optimal utilization of PMTCT services by adolescents. “ As their outcome of interest the authors chose being up to date with HIV counseling and testing during the antenatal period among 528 pregnant adolescents; and assessed clinical and demographic factors associated with this outcome. They reported that 75% of the adolescents who received Group focused antenatal care received test results compared to 42% of those receiving standard focused care. Factors associated with adolescent receipt of antenatal HCT included participation in Group ANC services, partner being tested for HIV, facility level, and earlier presentation for prenatal services.

While the findings are of interest, the manuscript had a number of concerns which detract from confidence in the findings. This included a number of proofing errors in the abstract and main manuscript with discrepancies between text results and the bivariate Table results; and some instances where reported prevalence ratios do not fall within the listed confidence intervals. In addition, the cascade of PMTCT services as described includes multiple other PMTCT outcomes of interest besides receipt of HCT services during the antenatal period, which were apparently were not assessed as part of the study analyses. If other parts of the PMTCT cascade were actually included in a composite outcome, this needs to be described/clarified in the methods text. And there is concern that the group of adolescents described are a very select group in terms of their high rates of health seeking behavior which makes generalizability of the results questionable.

Specific Comments

Abstract: The numbers in the abstract results for independent factors associated with optimal PMTCT do not exactly match those in Table 2 –e.g. partner tested results. In addition there appear to be typing errors where the PR does not fall within the CI range—partner tested as an example. These need to be harmonized.

Introduction/Background: Generally well written with good description of the PMTCT cascade. Sentence Line 59-60 is lacking several verbs and needs to be rewritten.

Study Design and Methods: The authors should clarify the rationale for choosing only one aspect of the PMTCT cascade—namely receipt of HCT results antenatally— as the primary outcome as opposed to a composite outcome, given the multiple other components of the full cascade including during the postnatal period.

The inclusion criteria as stated required that the adolescent mothers had received at least 18 months of care if HIV+ and 9 months if HIV negative, (p 5, line 100) and for the infants to be 9 months or less of age; so this is a very select group in terms of HIV infected adolescent mothers being already engaged in HIV health care. It also implies that HIV+ adolescent mothers would have known their status prior to the pregnancy if they had to have received 18 months of HIV care while their infants had to be aged 9 months or less, which again would be a very select group of adolescents who would have been aware of their HIV status prior to testing in ANC.

Statistical methods appear appropriate and sample size adequate to address the stated antenatal HCT outcome of interest.

Results:

Table 1 and text. It would be helpful to state the full range of age for the adolescents at the time of their enrolment in the study; as the vast majority >80%) were at least 18 years of age. Were most of them in their early 20’s or is upper age range just to age 19 years, and what is the youngest age of the pregnant adolescents included in the analyses?

Table 2 which presents the bivariate results needed a number of corrective edits including:

• While the text, line 176, p 11, for the bivariate analyses presents PR and CI for the G-ANC and F-ANC groups, this G-ANC and F-ANC data are not presented in Table 2.

• Re the age , column Yes, the percent >= 18 years is missing

• Re variable first ANC visit for first trimester there is an extra 1 typo in the confidence intervals.

• HIV test results there is a missing upper confidence limit

• In addition the text page lines 1, lines 177 -178 stated numbers for the PR and CI do not match the Table 2 numbers with regards to : 1st ANC visit in first or second trimester, or regarding partner tested, or for HCIV clinic.

Discussion: Generally well written. Of interest is the substantially higher uptake of PTMCT HCT for both the G-ANC and F-ANC among adolescents when compared to central Uganda. The authors should comment on what regional differences may account for this finding. Also of interest and surprising is the relatively high proportion of adolescents who made their first ANC visit in the first trimester for either the G-ANC (41.7%) or the F-ANC (42.1%). The authors could comment on how this early health seeking behavior among the adolescents was achieved! In addition to the limitations stated , it would also be important in terms of potential generalizability of findings that this was a select group of older adolescents with high health seeking behavior and therefore may not be applicable to younger pregnant adolescents in resource limited international settings.

In conclusion: some interesting findings on an important topic, but the authors need to address the concerns detailed above.

6. PLOS authors have the option to publish the peer review history of their article (what does this mean?). If published, this will include your full peer review and any attached files.

Reviewer #1: No

---

## [Author Response · Author response to Decision Letter 0]

6 May 2022

Responses to the reviewers’ comments

Academic editor

General comment

This is quite an interesting report on a strategy to improve PMTCT among adolescents. As an alternative to individual antenatal care, the authors are testing a group antenatal care. As an outcome, the authors use something they called “optimal utilization of PMTCT of HIV services”. This is a compositive outcome with important components the authors decided not to look at.

Response

Thank you for this summary of the paper. Our responses to each of the individual comments are provided below.

 According to the World Health Organization, the PMTCT strategy is a cascade of events that enable the identification of an HIV-infected pregnant woman and the prevention of HIV transmission to her infant. It comprises: attending antenatal care (ANC), HIV counseling and testing (HCT), and receiving HIV test results. If HIV negative, subsequent retesting in the third trimester, shortly after delivery, or within three months. If positive, antiretroviral therapy (ARVs) for the mother under option B plus (1), Nevirapine syrup (NVP), cotrimoxazole prophylaxis, and ARVs for the infant. Safe delivery, safe infant feeding, early infant diagnosis within 6 weeks, DNA PCR done 6 weeks after ceasing breastfeeding, serology at 18 months if DNA PCR is negative, and finally linkage into care and treatment if infant turns out positive (1) (2).

We therefore for HIV negative mothers considered first HIV counseling and testing before the third trimester, receipt of results and subsequent re-testing as recommended. For HIV positive mothers however, we looked at whether they were initiated on ART under option B plus. In addition, for the infants we asked if: there was safe infant delivery (delivery in a health facility), safe infant feeding (exclusive breast feeding for the first 6 months), infant testing, and receipt of results, nevirapine was given for six weeks, and cotrimoxazole was given until status was confirmed. Infant testing entailed early infant diagnosis (DNA PCR within 6 weeks), if the test turned positive, ART was initiated and the infant linked into care. We did not assess for the second DNA PCR test or serology as the infants had to be 9 months of age or less.

Comment #1 (Grammar and spelling errors)

There are so many avoidable English typos all over the manuscript. Please, do revise.

Response: This manuscript has been edited and the spelling errors are taken care of

Comment #2 (Abstract)

Please add in the abstract a short definition of this concept of “optimal utilization of PMTCT of HIV services”.

Response: This has been added on line 33 of the manuscript with track changes

Comment #3 (Study design and setting)

Line 85 in the study design: the rates presented there are yearly rates? Clarify these figures, please.

Response

 The most current data on the adolescent pregnancy rate in the Eastern region is from 2016. This along with the HIV pregnancy rate in 2020 have been specified on line 111 in the study design and setting section.

Comment #4 (Study participants and sampling)

 The eligibility criteria to be part of the study are quite unclear as the reviewer explains below. Did the women have to fulfill all the criteria in lines 99 to 101? Or just one criterion would be enough to participate in the study?

Response

 All the criteria listed had to be met; that is, the mothers had to: a) have sought care during the study period, b) have had completed at least 18 months of care if HIV positive, and 9 months of care if HIV negative c) have alive infants 9 months or less and d) provided written informed consent. This has been clarified in line 127.

Comment #5 (Sample size)

Sample size: is the effect size an absolute difference of the proportion? Or a relative difference. Clarify.

Response

 This was the absolute difference between the proportions and has been edited in line 136.

Comment #6 (Variables)

Line 124: How the exposure was measured here?

Response

At 2 health facilities (Jinja regional referral hospital and Budondo HCIV), F-ANC was the only mode of ANC available. However, at Iganga hospital and Bugembe HCIV, G-ANC was offered in addition to F-ANC. We checked the mode of ANC from their ANC card and/or the G-ANC register. We also asked the mothers whether they had received ANC care from any other health facilities and then assessed the ANC mode they had received. Line 155

Comment #7 (Data management and statistical analysis)

- Cite Epidata and Stata

- It is a bit problematic to use a mixed-effects model with just 4 clusters. Even with the sandwiched standard errors. I would suggest fitting fixed-effects models and keeping the health facility as 3 dummy indicators.

- Each one of the components of the “optimal utilization of PMTCT of HIV services” deserves to be analyzed. At least show the overall prevalence of these.

- Put the Ethical approval within the methods section

- There are so many abbreviations. Please provide a list

Response

• Epidata and stata have been cited lines 182-183

• The data have been re-analyzed using a fixed-effects model keeping the health facility as 3 dummy indicators. line 191.

• Each of the components of optimal utilization of PMTCT of HIV services has been analyzed using fixed effects regression modeling, and a prevalence ratio comparing G-ANC to F-ANC computed-line 191 

• The ethical approval has been moved to the methods section- lines 202 to 205.

• A list of abbreviations has been provided as supporting information.

Comment #8 (Results)

Be consistent with the proportions. In the tables 1 and 2 they are presented as percentages. But on the text lines 171 to 173, for example, they are something else.

Response: These have been edited to reflect percentages in lines 224 to 225.

Comment #9 (Table 1)

Table 1:

- Add the age descriptives described on line 163.

- At the HIV test result. What are the 1 and 2 rows?

- the mean travel time how was measured? This is not described in the methods.

- Please add as footnote how these p-values were computed

For the analysis of optimal utilization in line 169 to 173 I would request to add here the different components of this composite measure. There is an interest to see at what component the intervention did make a change.

Response

• The age descriptives have been added in table 1 as recommended.

• Row 1 is HIV positive mothers while row 2 is HIV negative mothers. These have been reflected in table 1.

• The mean clinic travel time was measured by self-report whereby the mothers were asked how much time it took them to get to the clinic where they received ANC. This has been added in the “Variables” section of the methods on lines 160 to 162.

• The P- values were computed using the Pearson’s Chi-square test for categorical predictors and the Wilcoxon rank-sum test for continuous ones. This has been added as a footnote to table 1.

• Each component of the outcome has been analyzed for ANC mode. This has been stratified by HIV test results as shown in table 2 and fig1. Prevalence ratios for the HIV positive mothers have not been computed due to the small numbers in this group.

Comment #10 (Table 2)

- Add one column for prevalence of “optimal utilization of PMTCT services”

- For the first ANC, please change the reference category

- Age in the column of “Yes” we have 242. Add its proportion.

In fact, the analysis of factors here is not that important. It is OK to adjust for these factors to have something like table 3. But I would prefer to have the all composite elements analysed to get their PR for the ANC type.

Response

- A column for the prevalence of “optimal utilization of PMTCT services” has been added to table 3

- The reference category for first ANC has been changed to “first trimester of ANC” as shown in table 3

- The proportion for age category ≥18 years has been added in table 3

- Each component of the outcome has been analyzed for ANC type. This has been stratified by HIV test results as shown in table 2 and fig1.

Comment #11 (Table 3)

Table 3 - this is a multivariable analysis. Correct the lone 193 as well. How this model came? How these covariates were chosen?

- Change the reference for the first ANC

Response

-This has been edited to ‘multivariable analysis’ as advised. 

-Prevalence ratios between individual predictors and the optimal utilization of PMTCT of HIV services at the bivariate level were assessed and those with a p-value less than or equal to 0.2 were considered for the multivariable analysis. A p-value less than or equal to 0.05 was considered statistically significant. The presence of interaction was assessed by using the log-likelihood test comparing the full and reduced models. Confounding variables were assessed by comparing prevalence ratios of unadjusted and adjusted models and those that caused a difference of at least 10% were considered confounders. We however did not find any significant interaction or confounding. The goodness of fit of the regression model to the data was assessed using the deviance test. 

-The reference category for first ANC has been changed to “first trimester of ANC” as shown in table 3

Reviewer #1

General Comments: This manuscript “Optimal utilization of Prevention of Mother –to Child Transmission of HIV services among adolescents under group versus focused antenatal care in Eastern Uganda” addresses a timely and challenging topic—how best to ensure uptake of PMTCT services by adolescents. The cross sectional study conducted in postpartum and immunization clinics, reports encouraging results with use of age and gestational grouping of antenatal services (“G-ANC) in terms of optimal utilization of PMTCT services by adolescents. “ As their outcome of interest the authors chose being up to date with HIV counseling and testing during the antenatal period among 528 pregnant adolescents; and assessed clinical and demographic factors associated with this outcome. They reported that 75% of the adolescents who received Group focused antenatal care received test results compared to 42% of those receiving standard focused care. Factors associated with adolescent receipt of antenatal HCT included participation in Group ANC services, partner being tested for HIV, facility level, and earlier presentation for prenatal services.

While the findings are of interest, the manuscript had a number of concerns which detract from confidence in the findings. This included a number of proofing errors in the abstract and main manuscript with discrepancies between text results and the bivariate Table results; and some instances where reported prevalence ratios do not fall within the listed confidence intervals. In addition, the cascade of PMTCT services as described includes multiple other PMTCT outcomes of interest besides receipt of HCT services during the antenatal period, which were apparently were not assessed as part of the study analyses. If other parts of the PMTCT cascade were actually included in a composite outcome, this needs to be described/clarified in the methods text. And there is concern that the group of adolescents described are a very select group in terms of their high rates of health seeking behavior which makes generalizability of the results questionable.

Response 

Thank you for this summary of the paper. Our responses to each of the individual comments are provided below.

According to the World Health Organization, the PMTCT strategy is a cascade of events that enable the identification of an HIV-infected pregnant woman and the prevention of HIV transmission to her infant. It comprises: attending antenatal care (ANC), HIV counseling and testing (HCT), and receiving HIV test results. If HIV negative, subsequent retesting in the third trimester, shortly after delivery, or within three months. If positive, antiretroviral therapy (ARVs) for the mother under option B plus (1), Nevirapine syrup (NVP), cotrimoxazole prophylaxis, and ARVs for the infant. Safe delivery, safe infant feeding, early infant diagnosis within 6 weeks, DNA PCR done 6 weeks after ceasing breastfeeding, serology at 18 months if DNA PCR is negative, and finally linkage into care and treatment if infant turns out positive (1) (2).

We therefore for HIV negative mothers considered first HIV counseling and testing before the third trimester, receipt of results and subsequent re-testing as recommended. For HIV positive mothers however, we looked at whether they were initiated on ART under option B plus. In addition, for the infants we asked if: there was safe infant delivery (delivery in a health facility), safe infant feeding (exclusive breast feeding for the first 6 months), infant testing, and receipt of results, nevirapine was given for six weeks, and cotrimoxazole was given until status was confirmed. Infant testing entailed early infant diagnosis (DNA PCR within 6 weeks), if the test turned positive, ART was initiated and the infant linked into care. We did not assess for the second DNA PCR test or serology as the infants had to be 9 months of age or less.

Comment #1 (Abstract)

The numbers in the abstract results for independent factors associated with optimal PMTCT do not exactly match those in Table 2 –e.g. partner tested results. In addition, there appear to be typing errors where the PR does not fall within the CI range—partner tested as an example. These need to be harmonized

Response

The results in the abstract have been harmonized to reflect those from the multivariable analysis reflected in table 3

Comment #2 (Introduction/Background)

Generally well written with good description of the PMTCT cascade. Sentence Line 59-60 is lacking several verbs and needs to be rewritten.

Response

This manuscript has been edited and the spelling errors are taken care of.

Comment #3 (Study participants and sampling)

The inclusion criteria as stated required that the adolescent mothers had received at least 18 months of care if HIV+ and 9 months if HIV negative, (p 5, line 100) and for the infants to be 9 months or less of age; so, this is a very select group in terms of HIV infected adolescent mothers being already engaged in HIV health care. It also implies that HIV+ adolescent mothers would have known their status prior to the pregnancy if they had to have received 18 months of HIV care while their infants had to be aged 9 months or less, which again would be a very select group of adolescents who would have been aware of their HIV status prior to testing in ANC.

Response

We recognize that this is a major limitation of this study as this group of older adolescents most likely have good health-seeking behavior and this affects the generalizability of the findings. We however believe that these findings can be generalized to older adolescents seeking health care at other health facilities.

Comment #4 (Data analysis)

Statistical methods appear appropriate and sample size adequate to address the stated antenatal HCT outcome of interest.

Response

We have re-analyzed the data using fixed effects regression modeling as advised by academic editor review.

Comment #5 (Table 1 and text)

It would be helpful to state the full range of age for the adolescents at the time of their enrolment in the study; as the vast majority >80%) were at least 18 years of age. Were most of them in their early 20’s or is upper age range just to age 19 years, and what is the youngest age of the pregnant adolescents included in the analyses?

Response

Table 1 has been edited to include the median age (18 years) and the 25th and 75th percentiles. The youngest adolescent was 14 years while the oldest was 19 years old- lines 212 to 213.

Comment #6 (Table 2):

Table 2 which presents the bivariate results needed a number of corrective edits including:

• While the text, line 176, p 11, for the bivariate analyses presents PR and CI for the G-ANC and F-ANC groups, this G-ANC and F-ANC data are not presented in Table 2.

• Re the age , column Yes, the percent >= 18 years is missing

• Re variable first ANC visit for first trimester there is an extra 1 typo in the confidence intervals.

• HIV test results there is a missing upper confidence limit

• In addition the text page lines 1, lines 177 -178 stated numbers for the PR and CI do not match the Table 2 numbers with regards to : 1st ANC visit in first or second trimester, or regarding partner tested, or for HCIV clinic.

Response

• The bivariate analysis by mode of ANC has been added to table 3, factors associated with optimal utilization of PMTCT of HIV services

• The proportion for age category ≥18 years has been added in table 3

• The confidence intervals for “first ANC” have been edited in table 3

• upper confidence limit has been added to HIV test results in table 3

• The numbers in the text and table 3 have been harmonized for trimester of first ANC visit, partner testing status.

Comment #7 (Discussion):

Generally well written. Of interest is the substantially higher uptake of PTMCT HCT for both the G-ANC and F-ANC among adolescents when compared to central Uganda. The authors should comment on what regional differences may account for this finding. Also of interest and surprising is the relatively high proportion of adolescents who made their first ANC visit in the first trimester for either the G-ANC (41.7%) or the F-ANC (42.1%). The authors could comment on how this early health seeking behavior among the adolescents was achieved! 

In addition to the limitations stated, it would also be important in terms of potential generalizability of findings that this was a select group of older adolescents with high health seeking behavior and therefore may not be applicable to younger pregnant adolescents in resource limited international settings.

Response

The higher optimal utilization of PMTCT of HIV services in the eastern region compared to the central observed could be because of the intensified efforts to promote maternal, child and adolescent health in the Busoga region by both the Ministry of Health and implementing partners. These include the Regional health integration to enhance services in the Busoga region (3), the government of Korea’s partnership with WHO and the Ministry of Health to support maternal, adolescent, and child health delivery in Busoga(4) among others (5, 6). As such, there has been collaboration with village health teams for mapping of pregnant adolescents; who are then linked to health care. This could also explain the high numbers of adolescent mothers attending ANC early i.e. in the first trimester. The observed difference could also be because the findings in central Uganda were from a study carried out in a national referral hospital, Mulago National Referral Hospital. There was likely difficulty in follow-up and as such poorer PMTCT service utilization.

We recognize that the generalizability of our findings is a major limitation of this study as this group of adolescents was older and also most likely had good health-seeking behavior. We however believe that these findings can be generalized to older adolescents seeking health care at health facilities. This has been adjusted in lines 355 to 357

---

## [Decision Letter · Decision Letter 1]

28 Jun 2022

PONE-D-21-31253R1Optimal utilization of Prevention of Mother-to-Child Transmission of HIV services among adolescents under group versus focused antenatal care in Eastern Uganda.PLOS ONE

Dear Dr. Akunzirwe,

Thank you for submitting your manuscript to PLOS ONE. After careful consideration, we feel that it has merit but does not fully meet PLOS ONE’s publication criteria as it currently stands. Therefore, we invite you to submit a revised version of the manuscript that addresses the points raised during the review process.

We look forward to receiving your revised manuscript.

Kind regards,

Orvalho Augusto, MD, MPH

Academic Editor

PLOS ONE

Journal Requirements:

Additional Editor Comments (if provided):

Thank you for the clarifying responses. There are still few more issues as the reviewer point below.

Reviewers' comments:

Reviewer's Responses to Questions

**Comments to the Author**

1. If the authors have adequately addressed your comments raised in a previous round of review and you feel that this manuscript is now acceptable for publication, you may indicate that here to bypass the “Comments to the Author” section, enter your conflict of interest statement in the “Confidential to Editor” section, and submit your "Accept" recommendation.

Reviewer #2: (No Response)

2. Is the manuscript technically sound, and do the data support the conclusions?

Reviewer #2: Yes

3. Has the statistical analysis been performed appropriately and rigorously? 

Reviewer #2: Yes

4. Have the authors made all data underlying the findings in their manuscript fully available?

Reviewer #2: Yes

5. Is the manuscript presented in an intelligible fashion and written in standard English?

Reviewer #2: Yes

6. Review Comments to the Author

Reviewer #2: The authors have responded satisfactorily to the previous reviewers' comments. My comments are mostly editorial.

1. Line 78. For clarity it would be helpful to define "safe delivery" and "safe infant feeding".

2. Line 93, place comma after "Although G-ANC",

3. Line 103. Better to say "This was a cross-sectional study..."

4. Line 105. Define HCIV. Many readers are not familiar with the Uganda health care system.

5. Line 142. Again define "safe delivery", "safe infant feeding".

6. Line 144. DNA PCR within 6 weeks. Is the process not DNA PCR within 6-8 weeks? Within 6 weeks implies this assessment being even within 2 weeks, which is incorrect.

7. Line 164. Need to describe the interviewers, their qualifications and training.

8. Line 166. A better term than "cross-examined" might be "cross-referenced", "correlated with", "checked with", "compared with".

9. Line 263/264. Need to provide evidence/reference for the statement that "...G-ANC was attributed to the

participatory nature of G-ANC resulting in a better relationship between health care workers."

10. Line 292. "Larger catchment areas and thus populations in health facilities result in difficulty in follow-up." This statement is not clear. If the authors are trying to say that larger catchment areas with larger populations make follow up more difficult, they need to provide evidence.

7. PLOS authors have the option to publish the peer review history of their article (what does this mean?). If published, this will include your full peer review and any attached files.

Reviewer #2: **Yes: **Godfrey Woelk

---

## [Author Response · Author response to Decision Letter 1]

11 Aug 2022

Responses to the reviewers’ comments

Academic editor

General comment

Response:

The reference list has been revised. References: 2,3,7,8, and 9 were previously retracted and have been updated with new literature. References: 4,6,8,15,17, 27,28, 29, 30, 31, 32, 33 were edited as they were incomplete. References 39-40 were replaced with more relevant references

Reviewer #2. The authors have responded satisfactorily to the previous reviewers' comments. My comments are mostly editorial.

Comment 1: Line 78. For clarity it would be helpful to define "safe delivery" and "safe infant feeding".

Response: Safe delivery, defined as delivery in a health facility by a qualified health worker has been added to line 78/79. Safe infant feeding defined as exclusive breastfeeding for six months, complementary feeding for up to 12 months for HIV-exposed infant, and 24 months for HIV-positive infants has been added to line 79/80.

Comment 2. Line 93, place comma after "Although G-ANC",

Response. A comma has been added as advised (line 95).

Comment 3. Line 103. Better to say "This was a cross-sectional study..."

Response. This has been edited as suggested (line 105).

Comment 4. Line 105. Define HCIV. Many readers are not familiar with the Uganda health care system.

Response. The definition of HCIV, a mini-hospital headed by a senior medical officer; has been added (line 107-108)

Comment 5. Line 142. Again define "safe delivery", "safe infant feeding".

Response. The definitions for safe delivery and safe infant feeding have been added on lines 78/79 and 79/80 respectively

Comment 6. Line 144. DNA PCR within 6 weeks. Is the process not DNA PCR within 6-8 weeks? Within 6 weeks implies this assessment being even within 2 weeks, which is incorrect.

Response. Thank you for this clarification. Yes, the first DNA PCR should be within 6-8 weeks. This has been edited on line 147

Comment 7. Line 164. Need to describe the interviewers, their qualifications and training.

Response. We trained 6 interviewers (diploma-level nurses) for 4 days on the questionnaire, data collection procedures, and sampling methods. This has been added on line 162

Comment 8. Line 166. A better term than "cross-examined" might be "cross-referenced", "correlated with", "checked with", "compared with".

Response. The term has been changed to cross-referenced (line 170)

Comment 9. Line 263/264. Need to provide evidence/reference for the statement that "...G-ANC was attributed to the participatory nature of G-ANC resulting in a better relationship between health care workers."

Response. We think that the observed better optimal utilization of PMTCT of HIV services under G-ANC could be attributed to improved client satisfaction and prenatal service utilization. The improved client satisfaction has been attributed to the participatory nature of G-ANC which improves relationships between health care workers and mothers as well as among the mothers (Line 267-269). References to support these statements have been added to line 270.

Comment 10. Line 292. "Larger catchment areas and thus populations in health facilities result in difficulty in follow-up." This statement is not clear. If the authors are trying to say that larger catchment areas with larger populations make follow-up more difficult, they need to provide evidence.

Response. 

The statement, line 297/298, has been edited to ‘Larger catchment areas with larger populations likely result in high workloads which affects the quality of patient care’. References to support this statement have been added to line 298/299

---

## [Decision Letter · Decision Letter 2]

26 Sep 2022

Optimal utilization of Prevention of Mother-to-Child Transmission of HIV services among adolescents under group versus focused antenatal care in Eastern Uganda.

PONE-D-21-31253R2

Dear Dr. Akunzirwe,

We’re pleased to inform you that your manuscript has been judged scientifically suitable for publication and will be formally accepted for publication once it meets all outstanding technical requirements.

Kind regards,

Orvalho Augusto, MD, MPH

Academic Editor

PLOS ONE

Additional Editor Comments (optional):

Sadly, as the authors note Uganda among other many African nations have the highest adolescent motherhood in the world. This is an important work to help improve the health of adolescents mothers in Sub-Saharan Africa. This manuscript has improved substantially. However, before publication some issues need to be addressed.

1. Table 1 - I particularly dislike seeing p-values in this table. The assessment of balance between the groups of exposure should be done with substantial grounds rather than the solely p-values statistical grounds. Anyway, for marital status, education and first ANC visit a single p-value should be reported for each of those variables. I understand that the authors used a procedure that only provides cluster adjusted t-statistic. However, if indeed only t-tests are provided we should have k - 1 (categories) p-values. For example, in the marital status there should be 2 p-values not 3 because one of the categories becomes a reference. Therefore, I urge the authors to remove the p-values at all from this table. In fact, you never use that information in the paragraph dependent of this table.

2. Table 2:

- Put below the table the meaning of PR (prevalence ratio?)

- Add information on how the PR are computed, These are compute from Poisson regression with robust standard errors, right?

- And these all are unadjusted estimates

3. Table 3:- Put below the table the meaning of PR

In addition to these please, address the reviewers comments.

Reviewers' comments:

Reviewer's Responses to Questions

**Comments to the Author**

1. If the authors have adequately addressed your comments raised in a previous round of review and you feel that this manuscript is now acceptable for publication, you may indicate that here to bypass the “Comments to the Author” section, enter your conflict of interest statement in the “Confidential to Editor” section, and submit your "Accept" recommendation.

Reviewer #2: All comments have been addressed

2. Is the manuscript technically sound, and do the data support the conclusions?

Reviewer #2: Yes

3. Has the statistical analysis been performed appropriately and rigorously? 

Reviewer #2: Yes

4. Have the authors made all data underlying the findings in their manuscript fully available?

Reviewer #2: Yes

5. Is the manuscript presented in an intelligible fashion and written in standard English?

Reviewer #2: No

6. Review Comments to the Author

Reviewer #2: The manuscript has been revised to an acceptable standard. However, I have a few edit comments.

1. Abstract; line 46. This is a very strong recommendation (implement widely) based on a single cross-sectional study. I suggest the authors should moderate this recommendation, by for example, the MOH consider widely implementing G-ANC.

2. Introduction; line 72. The term "HIV-positive" rather than "HIV-infected" mothers is preferred to reduce potential for stigma. Similar comment in line 84

3. Line 77. anti-retroviral therapy, not Anti-retroviral therapy.

4. Line 87. Ministry of Health should be capitalized.

5. Lines 152, 153, 156. No need for the word "like" when describing the variables for which data were collected.

6. Line 234, "Five (83.3%) of the ....". The percentage is misleading as the total of 6 is small. Better to report 5 of 6 mothers. Similarly, in line 236, it is more accurate to report 7 of the 12 mothers, rather than present a percent.

7. PLOS authors have the option to publish the peer review history of their article (what does this mean?). If published, this will include your full peer review and any attached files.

Reviewer #2: No

---

## [Editor Report · Acceptance letter]

20 Oct 2022

PONE-D-21-31253R2 

Optimal utilization of Prevention of Mother-to-Child Transmission of HIV services among adolescents under group versus focused antenatal care in Eastern Uganda. 

Dear Dr. Akunzirwe:

I'm pleased to inform you that your manuscript has been deemed suitable for publication in PLOS ONE. Congratulations! Your manuscript is now with our production department. 

Kind regards, 

on behalf of

Dr. Orvalho Augusto 

Academic Editor

PLOS ONE